# The cumulative disadvantage of unemployment: Longitudinal evidence across gender and age at first unemployment in Germany

Anna Manzoni [1]◉*, Irma Mooi-Reci[2]◉

1 Department of Sociology and Anthropology, North Carolina State University, Raleigh, North Carolina, United States of America, 2 School of Social and Political Sciences, Faculty of Arts, The University of Melbourne, Parkville, Victoria, Australia

◉ These authors contributed equally to this work.
* anna.manzoni@gmail.com

**Data Availability Statement:** We are using secondary data which can be obtained from DIW Berlin. See https://www.diw.de/en/diw_02.c.222829.en/access_and_ordering.html.

## Abstract

Unemployment is an important predictor of one's future employment success. Yet, much about the endurance of unemployment effects on workers' careers remains poorly understood. Our study complements this knowledge gap by examining the rate of recovery in the quality of careers following an unemployment spell among a representative sample of previously unemployed workers with different socio-demographic characteristics in Germany. We apply a new dynamic measure that quantifies the quality of binary sequences, distinguishing between "good" (i.e., employment) from "bad" labor force status activities (i.e., unemployment and inactivity). We use longitudinal data from the German Socio-Economic Panel (GSOEP) before the Great Financial Recession over the period 1984–2005 and deploy a series of hybrid models that control for unobserved heterogeneity. We find a nonlinear recovery process after unemployment across gender and age groups. That is, after a period of recovery, career quality worsens. Least impacted are men experiencing unemployment when aged between 25–34 years, while men 55–66 have rather stable, though stronger, penalties. Furthermore, we find that recovery processes are contingent upon *when* respondents experience unemployment.

## Introduction

Scholars have long been interested in the role that unemployment plays for workers' labor market outcomes and achievements. We know much about the negative consequences of unemployment on re-employment chances [1–5] and the imposed wage penalties [6–11]. We also know that negative effects attached to unemployment spread to the next generation impacting negatively on children's educational achievements and their employment outcomes [12–15]. Furthermore, existing studies have revealed that unemployment effects vary depending on age, gender and economic fluctuations [10, 16] and that much of the size and endurance

**Funding:** The author(s) received no specific funding for this work.

**Competing interests:** The authors have declared that no competing interests exist.

of these effects is shaped by the social context and subjective perceptions around unemployment [17–19].

A key theoretical notion reflected in these studies is the concept of cumulative advantage/disadvantage (CAD), according to which the material, social and skill deprivations attached to (disruptive events such as) unemployment build on themselves in self-reinforcing cycles to influence negatively the economic and labor force status of affected workers (for a review see, 20). In analyzing employment outcomes after unemployment, a central tenet of the CAD perspective is therefore that labor market inequalities accumulate over time. However, sociologists have observed that, under certain circumstances, cumulative inequality processes in the labor market can be reversed. For example, simulation analyses in the United States and West Germany have indicated that supportive unemployment benefit institutions can offset up to 20% of the cumulative disadvantages associated with unemployment for German workers [6]. Based on a set of reforms in the Dutch unemployment benefit system during the 1980s and 1990s, supporting evidence about the offsetting role of generous institutions was also found for the Netherlands [20, 21]. These findings imply that cumulative inequality processes attached to unemployment could be offset by supportive benefit institutions and efficient job matching, and that, at least theoretically, full recovery from previous spells of unemployment should be possible over the course of one's career.

Theoretically, evidence of such recovery process would contrast with the traditional formulation of the CAD perspective that assumes early disadvantages to only grow larger over time. Empirically, knowledge about the endurance of cumulative inequality processes following unemployment is necessary to understand how labor market inequalities play out over time, who is most impacted, and when. Yet, despite three decades of progress on unemployment scarring, we know very little about how *fast* workers' careers recover from unemployment—if at all—or how such recovery processes differ by gender and age groups. Nor do we know if recovery patterns follow a linear or a non-linear trend, or if the speed of recovery varies by gender and over the life course. This is important if we want to promote change. Previous studies on the effects of unemployment have usually focused on one unemployment dimension (i.e., occurrence or spell of previous unemployment) on outcomes such as earnings, employment, health or fertility of displaced workers [e.g., 1–5]. While valuable, this approach misses labor force status changes over the course of one's career, and how these dynamics in turn shape people's employment trajectories and their level of career success in the future. Sequence analytical approaches that measure labor force status changes in the form of career 'turbulence' or 'complexity' have been developed recently, but these measure the 'volatility' of labor force changes rather than the quality of these changes [e.g., 22].

Our study addresses these gaps by focusing on the level of career quality and the speed of recovery following an initial spell of unemployment. Drawing on recent advances by Manzoni and Mooi-Reci [23, 24], it quantifies the quality of binary sequences not only across gender, but also across different age groups which proxy different life stages, distinguishing between "good" (i.e., employment) from "bad" labor force status activities (i.e., unemployment and inactivity). The advantage of this newly developed measure is that it captures the volatility of labor force trajectories and their evolution since the occurrence of an initial unemployment spell. In addition, and more importantly, it quantifies the quality of labor force status trajectories in a dynamic, holistic and multidimensional way such that the measure decreases with unfavorable activities (such as unemployment and inactivity) and increases with favorable employment experiences. As such, this is the first sequence-based measure that quantifies the overall quality of labor force outcomes, and thus the *career recovery* from unemployment, allowing us to test directly and empirically CAD-specific formulations. From a policy

perspective, understanding which groups are more vulnerable and when helps design more effective public policy.

We use monthly longitudinal data from the German Socio-Economic Panel (GSOEP) from a representative sample of men and women who experienced unemployment between the ages 18 and 64 to create a holistic longitudinal measure for career quality [25]. We focus on Germany because of its generous unemployment benefit system, which has been found to counteract enduring effects of unemployment [6]. To avoid capturing processes that could be related to the Great Financial Recession in the second half of the 2000s, our analyses span the period 1984–2005. We focus on the diversity of labor market outcomes following an initial spell of unemployment and model differences in the labor force status outcomes across gender and age groups through a series of hybrid panel models [26]. After accounting for unobserved heterogeneity, we find that among those who experienced unemployment, labor force status outcomes are skewed along the lines of age and gender. In this way, we provide direct and complementary evidence about the endurance of inequality processes in the labor market. To the best of our knowledge, our study is one of the few [23, 24] attempting to quantify CAD processes following unemployment over one's careers and across different socio-demographic groups.

## Empirical evidence on unemployment scarring

Recurrent unemployment has continuously challenged researchers with the question: does current unemployment breed future unemployment? Research findings remain mixed. Some studies find no association between early and future unemployment experiences [1, 27, 28]. For example, using data from the United States on monthly labor market histories from the National Longitudinal Surveys of Young Men 1969–1971, Heckman and Borjas [28] and later Ellwood [27] find no association between past and future unemployment. Both studies find that (relatively short) unemployment spells of maximum 30 weeks have virtually no impact on recurrent unemployment spells. Likewise, Corcoran and Hill [1] investigated unemployment state dependence during later ages using data from the Panel Study of Income Dynamics (PSID) in the United States. When controlling for differences in individual or environment characteristics, the association between past and later unemployment remained. However, when controlling for errors in the data collection procedures, the association disappeared, leading to the conclusion that the earlier findings were related to unmeasured constant personal differences that influence the likelihood to re-experience unemployment [1].

Yet, other studies do find a significant relationship between past and future unemployment experiences. Using the same PSID data and focusing on male household heads between 25 and 54 years of age, DiPrete [29] shows a positive relationship between early and future unemployment spells, which varies strongly over the life—and the business cycles. The study reveals that unemployment state dependence varies across age and ethnicity group, while it exacerbates over different economic cycles. Findings from this study suggest that unemployment state dependence effects are dynamic and skewed along lines of race, age and business cycles. Additional research from Omori [16] supports the findings about dynamic unemployment state dependence effects. Using a sample of young men from the National Longitudinal Survey of Labor Market Experience Youth Study (NLSY) 1979–1987, he demonstrates that the duration of previous unemployment lengthens the expected duration of unemployment. According to Omori [16], this relationship is stronger and greater during economic upturns because signaling effects attached to stigma around unemployment emerge when proportionately fewer are unemployed. More recently, Torgovitsky [30] finds that unemployment accounts for at least 30–40% of the four-month persistence in unemployment among high school educated men.

Advances in the literature have been also made in Europe, where growing research has demonstrated the existence of unemployment state dependence effects. In the United Kingdom, research using data from the British Household Panel Survey (BHPS) between 1991–1997 has found strong state dependence effects [31]. For Dutch workers, unemployment repetition causes the highest scarring effects among men while for women the first unemployment spell inflicts the deepest scars followed by the recency of unemployment [10, 32]. Among German workers, the size of unemployment scarring is related to the duration of unemployment spells. Specifically, each additional day of unemployment during the first eight years on the labor market has been found to increase unemployment in the following 16 years by half a day [33].

Overall, while unemployment state dependence has been an active area of research, several issues remain not fully assessed. First, the majority of early research on scarring has focused on male workers, lacking emphasis on the effects of unemployment for women's careers. Although a growing number of studies are including women into their analyses, they have been limited to women's wages or earnings profiles and less on their career outcomes following unemployment. Our study addresses this gap by studying the career scarring attached to unemployment and how this varies across gender and age groups. Second, despite three decades of progress, a wide range of findings about unemployment state dependence and wage scars are based on single unemployment dimensions (i.e., some studies focus on duration others on the occurrence etc.), thus ignoring the added combination of the occurrence, duration and frequency of unemployment for one's career progress. Our study contributes to this limitation by means of an innovative measure that allows us to investigate the combined effect of all unemployment dimensions on the rate of career quality following initial unemployment. Finally, most of previous research has used relatively short observation periods, which provide a limited view about recovery processes. Our study, instead, takes advantage of the long observation periods in the GSOEP data to track up to almost 23 years of a worker's career to provide more accurate estimates about the rate of career recovery after unemployment.

## Background and theoretical framework

Unemployment is an important predictor of one's future career achievement. Its incidence and duration determine both the type and the quality of subsequent jobs, which influence career success and achievements in the future [4, 34–36]. Within a CAD perspective, unemployment amplifies inequality processes when employment opportunities deteriorate over one's career. Such trends could reflect: (1) depreciating human capital, (2) adverse signaling effects or (3) the rising polarization in the labor market with the majority of previously unemployed workers in non-standard and contingent forms of employment. In the following, we will describe each of these mechanisms.

### Accumulating labor force disparities

Within a CAD perspective, labor market inequalities attached to unemployment have their origins in the human capital depreciation, which amplifies over extended or repeated spells of unemployment [37, 38]. In human capital theory [37], the extent of future labor force penalties will depend on the loss of specific skills and other benefits tied to a previous occupation. That is, the longer the unemployment spell over the course of the career, the larger the depreciation of the specific skills and the greater the relative labor force status penalties in the future. Even in the case of re-employment, those who were previously unemployed will experience more difficulties to update and upgrade their lost human capital. Upgrading of specific skills is usually acquired on the job. However, to minimize the risk of no returns, employers will be more

often inclined to offer training opportunities to those higher educated or more tenured work-ers who have a regular and continuing position. Quantitative studies in the U.S. and Europe find empirical evidence that unemployment duration is positively related with skill erosion and depreciation of human capital [39, 40]. Thus, within a human capital perspective, the length of a previous spell is a central mechanism driving future career scars.

Another mechanism that could lead to group differences in labor force outcomes over the course of the career is adverse signaling. Originating from the signaling theory [41], the idea is that employers have imperfect information about the quality and qualifications of job appli-cants and therefore use cues or signals from job applicants' resumes to infer about their poten-tial productivity [41]. It follows that alterable attributes (i.e., attributes that are thought to be influenced by the job applicant) such as high education or stable employment histories are seen as potential success factors, while unemployment or non-standard employment histories raise suspicion about one's motivation and commitment [2, 19, 39, 41, 42]. For an employer, recent spells of unemployment are more directly related with one's productivity at the time of application and will thereby raise more severe red flags than spells experienced far back in the past. Though not directly related to unemployment, recent empirical research in the United States finds that workers with recent histories of non-standard employment face more severe penalties at the hiring interface than those otherwise [19]. Much of these penalties were linked to employers' preconceptions about the productive capabilities of workers in non-standard employment contracts, which are inferior to more regular and ongoing forms of employment. Thus, from a signaling perspective, the recency and the length of a previous unemployment spell are two major factors that determine the level of career scarring attached to previous unemployment.

Finally, expectations about group differences in labor force trajectories are framed within the labor market segmentation theory [38, 43]. The segmentation theory depicts the labor mar-ket as divided between "good" jobs that offer wage and career progress and "bad" jobs that are associated with job insecurity and limited opportunities for career progression. Previously unemployed with long and repeated spells of unemployment are expected to end up in such "bad" jobs that will negatively affect workers' long-term employment prospects by increasing the chances of recurring unemployment. Indeed, empirical evidence based on longitudinal data among unemployment insurance (UI) recipients demonstrates that previously unem-ployed individuals are more likely to end up in low-status and low-paying jobs [6, 21], which we think reflect jobs that are located in the secondary labor markets. Thus, from a labor market segmentation perspective, the repetition or frequency of previous unemployment is highly cor-related with precarious employment in the future, which in turn determines the level of future career scars. Overall, what we learn from these theories is that the duration, recency and fre-quency are three unemployment dimensions that drive scarring. Combined together these dimensions predict a downward moving career level. Hence, we expect that:

*Hypothesis 1A*: *Differences in the career quality between the pre and post-unemployment peri-ods will widen over one's career.*

## Narrowing labor force disparities

Cumulative inequality processes in the labor market may be reversed. Stratification research demonstrates, for example, that employment can counteract effects of unemployment over the course of one's career, through the accumulation of new skills and work [29, 44]. Generous institutional resources are also found to reverse labor market misfortunes [6, 44]. This

resonates with accounts that generous unemployment benefit institutions enhance workers' bargaining power by providing time to search, select and apply for jobs that best match with workers' skills and qualities. It follows that compensating workers for lost earnings and relieving them of the financial pressures during unemployment will avoid a downward adjustment of wages and job quality [6]. Consequently, this will raise the quality of selected jobs and increase the matching quality, thus helping maintain the acquired human capital and skills. In the context of re-employment, efficient job matching will enable workers to reveal their true abilities and productive capacities and will contribute to stable employment histories and more favorable career trajectories over time.

The German unemployment benefit system is considered generous in the literature [6, 44]. Prior to the Hartz IV reforms that passed into law in 2005, the unemployment benefit system consisted of three layers Hochmuth [45]. First, those who worked for at least 12 months in the three years preceding unemployment were guaranteed incomes between 6 and 32 months through the so-called "Arbeitslosengeld" Schmieder, von Wachter and Bender [46]. Short-term unemployed (<12 months) without children received benefits with a replacement rate of 60% and those with children received 67% of their previous net wages. Second, upon exhaustion of unemployment benefits, long-term unemployed received the so-called "Arbeitslosenhilfe" that was set at a fixed replacement rate of 53% and 57% for respectively those without and with children with no limited duration. Third, social assistance, or the so-called "Sozialhilfe", offered a remedy for those with insufficiently high levels of replacement rates, and for those who could not qualify for either the first or the second layer of the unemployment benefits due to short or lacking employment histories. Social assistance was (and continues to be) means tested, with no specific employment history requirements, offered at a fixed rate. The generosity of the German benefit system between 1985–2005 has to be viewed in the context of its economic performance. Especially during the 1990s and until 2005, Germany's economic growth was slower than that of other countries in the Euro Area [47]. The lagging economic growth was coupled with relatively high unemployment rates that varied between 8% in 1985 and 11% in 2005. A contracting labor market that involves more competition for jobs suggests that disadvantaged groups such as younger workers and women would be most impacted.

Empirical evidence among a sample of German and U.S. unemployed workers shows that generous unemployment benefits buffer the negative effects of unemployment on both wage outcomes and employment stability, with much stronger and larger buffering effects in the German context [6]. Mooi-Reci [10] also confirmed the buffering role of institutions in a study measuring the impact of cuts in the generosity of unemployment benefits in the Netherlands, finding that restricting resources of previously unemployed leads to both faster transitions to poor quality jobs and higher likelihood of recurrent unemployment. More recently, Schmieder, von Wachter and Bender [48] have examined the effect of extending the duration of UI benefits on non-employment duration; they reported lower subsequent probability to be not active in the labor market and wage gains for individuals receiving unemployment benefit for longer durations. These findings suggest that a generous UI benefit system can lead to substantial wage recovery and improve future employment stability among unemployed workers. Hence, based on the buffering role of institutions, we expect unemployment effects to evolve opposite to directions of Hypothesis 1A. Specifically, we expect that:

*Hypothesis 1B*: *Differences in the career quality between the pre and post-unemployment periods will narrow over one's career.*

## Evolution of career disadvantage across gender and age

Cumulative inequality processes related to unemployment are not uniform but vary contingent upon context [16, 49, 50]. Recent quantitative evidence among a Dutch sample of working men and women demonstrates that unemployment effects play out differently by gender, age and economic fluctuations [10]. Among others, Mooi-Reci and Ganzeboom [10] show that particularly men and those unemployed at older ages experience stronger unemployment scars compared to women and to individuals experiencing unemployment at earlier ages. Loss of human and social capital can in part explain this finding. For instance, older workers' skills typically lose their significance more rapidly in a vast changing business environment, while younger workers may have skills suited to newly created jobs that older workers lack [51]. Thus, older workers would be less likely to be re-hired, and thus have longer unemployment spells. Furthermore, older workers undertake a more selective job search than younger workers as their experience, skills, and networks have typically become tied to specific occupations, industries, or sectors. This selective job search process is predictive of longer unemployment spells and lower likelihood of being hired.

Another factor leading to gender and age differences among the previously unemployed could relate to the notion of the "ideal worker", which takes its origin from the status characteristic theory [39]. The theory builds on the idea that cultural expectations about men's and women's family work influence employers' evaluations about the competence and commitment of different gender and age groups. It follows that expected fragmentations of full-time employment due to child-rearing and bearing, influence employers' expectations about the level of one's performance and commitment at work, which in turn negatively affect women's and especially mothers' wage outcomes [39]. This implies that when women's work patterns and interruptions are consistent with the established gender role expectations, such interruptions will be less scrutinized by employers and should carry less of a negative hiring signal. Hence, women's likelihood of reemployment in the post-unemployment period should not decrease. Relatedly, if social norms about gender roles determine employers' hiring decisions, then career inequalities should be more pronounced among men who deviate from such expected gender roles. Particularly, men deviating from the established traditional breadwinner model are expected to be judged more harshly, deemed incompetent, less committed and not motivated to work hard. In turn, this would predict lower reemployment chances and thus slower career recovery post unemployment. Overall, we hypothesize that if the notion of the "ideal worker" serves as a mechanism that activates age and gender stereotypes, then:

*Hypothesis 2A*: *Differences in the career quality between the pre and post-unemployment periods will be wider for men than women.*

*Hypothesis 2B*: *Differences in the career quality between the pre and post-unemployment periods will be wider for men experiencing unemployment around the child-bearing (25–35) and child rearing (36–45) ages than women of the same age.*

*Hypothesis 3*: *Differences in the career quality between the pre and post-unemployment periods will be wider for those experiencing unemployment at older (36–64) than at younger ages (18–35), independent of their gender.*

## Data, measures, and analytical strategy

### The German Socio-Economic Panel data

We use data from the German Socio-Economic Panel (GSOEP) collected between 1984 and 2005 for West German respondents between 18 and 64 years of age [25]. The GSOEP data

offers a wide range of information on private households and detailed retrospective information about participants' labor force status. Relevant to this study, is the calendar data information about the labor market status of the respondent in the previous calendar year allowing us to track individual employment careers on a monthly basis. The retrospective information gathered at the first interview reveals whether employment spells were ongoing when the first interview began. Exploiting this information, we select respondents who: (i) experienced unemployment sometime over the period 1984–2005; (ii) were observed before they experienced unemployment (that is, they were not already unemployed at their first observation); and (ii) were at least 18 years or older at their first interview, but who retrospectively reported to have been continuously employed (either with the same or another employer). In this way, we exclude the possibility that a respondent experienced unemployment prior to the interview. This selection leads to a sample of 93,354 person-month observations across 1,141 women and men.

Next, we consider the first unemployment incidence in one's career, and quantify how the quality of career trajectories evolves in the period following that initial unemployment spell. Career sequences could take any number of paths after a first unemployment experience: workers could remain unemployed, withdraw permanently from the labor market, be continuously employed thereafter, be partly employed and experience other spells of unemployment thereafter or be partly employed and partly inactive or retired thereafter. We explore the differential effects of unemployment by sex and age at first occurrence.

## Measures

**Dependent variable: Career quality.** Our dependent variable is *career quality*, a measure developed by Manzoni and Mooi-Reci [23]. This is a time-varying measure that captures the quality of career sequences starting from a first unemployment experience up until $t$. It is based on a non-alignment sequence technique [22] and relies on the assumption that a career consists of a chain of different labor market states (i.e., sequences) that vary across workers. We measure these chains of labor market events on a monthly basis and distinguish between four major labor market states that can occur over the course of one's career: employment (E), unemployment (U), inactivity (I), and retirement (R). To illustrate how a typical labor market sequence may look on a monthly basis, consider sequence $x = EEUUIEE$. This sequence represents an observed hypothetical career of 7 months that starts with two months of employment, followed by a two-month spell of unemployment, a month of inactivity and ends with two months of employment. Transitions in and out of work characterize this sequence and the career quality measure infers that less successful careers are characterized by more frequent and longer durations of unemployment or inactivity spells that detach workers from labor market activities. The career quality measure is based on the assumption that the timing and recency of unemployment spells will determine how fast one recovers from unemployment. That is, the more frequent E-observations after a period of unemployment and the more recent these are relative to the total sequence length, the faster the recovery will be. Our career quality measure combines these common features of career sequences into a continuous measure that: a) has a fixed range of [0, 1]; b) increases with the number of successes (i.e., employment); and c) decreases with the number of failures (i.e., unemployment or inactivity experiences). Further details about the features and the mathematical derivations of the career quality measure are summarized in Manzoni and Mooi-Reci [23].

**Independent variables.** To investigate the evolution of careers after the occurrence of an initial episode of unemployment, we create and use dummy variables counting: (a) the trimester before the first unemployment, (b) the month of first unemployment, and (c) 90 trimester

dummies capturing the period after the first unemployment in which a respondent was observed in any of the labor market statuses.

We also control for socio-demographic characteristics that are endogenous to unemployment. These include continuous variables capturing *age* and *age squared*. *Education*, defined as the highest attained educational level before any labor force status change, is specified as a categorical variable with low education level as the reference category and three additional categories for low intermediate, high intermediate, and high education. We create a dummy variable for those who miss a valid education level at the survey time. We also include a variable for the *Gross Domestic Product* (GDP) per capita as proxy of economic growth in a specific year. Finally, to guard against the possibility that observed career outcomes are driven by career fluctuations that existed prior to the first unemployment (e.g., due to periods of inactivity), we include a variable specifying the *career quality before the first unemployment experience* in our models. Additionally, we construct a categorical variable for age and look separately at the following age groups: 18–24; 25–35; 36–45; 46–54; and 55–64 years old. Table 1 reports the mean and standard deviation of our variables.

**Method.**   To approximate a fixed-effect model that deals with time-constant observed and unobserved factors influencing over-time changes in career quality, we estimate a hybrid model [26]. Combining strengths from both random—and fixed-effect models, hybrid models offer an alternative when key independent variables are constant and thus dropped from a fixed-effect analysis. In essence, the hybrid model fits a fixed effect model by including both cluster-specific means of time-varying individual characteristics and deviations from such cluster-specific means. The model takes the following linear form:

$$C_{it} = \alpha + \boldsymbol{\beta}_0(x_{it} - \bar{x}_i) + \boldsymbol{\beta}_1\bar{x}_i + \mu_i + \varepsilon_{it} \tag{1}$$

where $C_{it}$ denotes the measure for career quality of worker $i$ at time $t$ (covering the period 1984 to 2005), $\boldsymbol{\beta}_0$ denotes a vector of coefficients associated with deviations from cluster-specific means expected to be associated with variations in one's career quality (including education and GDP), while $\boldsymbol{\beta}_1$ is a vector that captures the coefficients associated with the cluster-specific means. Finally, $\mu_i$ refers to the individual specific error (i.e., level 2 error) while $\varepsilon_{it}$ refers to the level 1 error. To follow the trend of recovery from an initial spell of unemployment, we extend Eq (1) by including dummies for each trimester since the initial unemployment and a separate dummy for the month of first unemployment. Similar to Eq 1, time

**Table 1. Descriptive statistics of study measures.**

|  | Men | | Women | |
|---|---|---|---|---|
|  | *Mean* | *SD* | *Mean* | *SD* |
| Career quality | 0.59 | 0.42 | 0.46 | 0.4 |
| Age | 40 | 13.6 | 37 | 11.3 |
| Career quality before unemployment | 0.76 | 0.37 | 0.62 | 0.4 |
| GDP | 3.45 | 2.21 | 3.47 | 2.23 |
| Education |  |  |  |  |
| *low* | 71.01% |  | 59.50% |  |
| *low intermediate* | 14.62% |  | 30.81% |  |
| *high intermediate* | 5.98% |  | 5.40% |  |
| *high* | 7.59% |  | 3.83% |  |
| *missing* | 0.81% |  | 0.45% |  |
| N observations | 41,677 |  | 51,677 |  |
| N workers | 536 |  | 605 |  |

dummies are demeaned from the overall time cluster mean and a trimester specific cluster-mean is included in the model. As noted by Allison [26] and Schunk and Perales [52], interpretations of this model rely on coefficients that are associated with deviations from cluster specific means. These coefficients reflect how on average a within-cluster change in individual characteristics $x_{it}$ is associated with a within-cluster change in the career quality variable.

## Results

### Gender differences in post-unemployment career quality

The first two columns in S1 Table show the results of the hybrid models that test our first two hypotheses that predicted differences in the career quality pre and post-unemployment to either widen (Hypothesis 1a) or narrow (Hypothesis 1b) over the course of one's career. To ease the interpretation, Fig 1 plots the coefficients for the effects of trimester dummies on women's and men's career quality from S1 Table. These coefficients indicate the estimated differences in the career quality in each trimester following unemployment relative to the trimester before unemployment (the reference category), all else equal. It is important to note that coefficient estimates that come closer to 0 indicate no or small differences in the level of career quality between the pre and post-unemployment, pointing to narrowing career penalties, or what we call a 'career recovery'. Coefficient estimates that are negative and further away from 0 indicate what we call a 'career penalty'.

Fig 1 clearly shows that differences in career quality compared to before unemployment are widest at the time of unemployment ($b$ = -0.609 for women and $b$ = -0.758 for men), after which differences in the career quality become gradually smaller among both men and women. This gradual career recovery is likely to resonate reemployment effects which boost recovery processes among women and men. About 6 years after first unemployment differences in women's and men's career quality start to diverge. Specifically, after about 6 years (from the 25th trimester and onwards), women experience a widening gap in their career quality relative to the trimester prior unemployment, which translates into a gradual drop in their career quality. This drop may reflect repeated unemployment spells related to childbearing and rearing periods. In contrast, men's recovery process continues to progress. Interestingly, for women, the career quality never returns to the pre-unemployment level. For men, it takes almost 15 years to reach the level of career quality of the trimester prior to initial unemployment. Altogether, these results indicate that differences in career quality pre and post-unemployment follow a non-linear trend, which is a novel finding. For men, these initial results are consistent with Hypothesis 1b. For women, the picture is more complicated. That is, the first 6 years of following initial unemployment reveal narrowing career penalties, lending support to Hypothesis 1b, but these effects are reverted and widened 6 years afterwards, lending support to Hypothesis 1a.

The last column in S1 Table examines whether the effect of unemployment on the level of career quality is moderated by gender (Hypothesis 2a) by testing differences in the time effects between men and women. We find statistically significant gender differences in career quality in the first 6 months following unemployment. Coefficients for the subsequent trimesters following initial unemployment up until about 6 years since unemployment are slightly smaller for women than men, but these gender differences in the speed of recovery are not statistically significant. Gender differences remain statistically insignificant afterwards, although while men keep recovering, women's career quality seems to worsen, enlarging the gender gap. This suggests that unemployment leaves a "chronic" and significant penalty in the careers of women, more so than of men. Altogether the results are partially consistent with Hypothesis 2a.

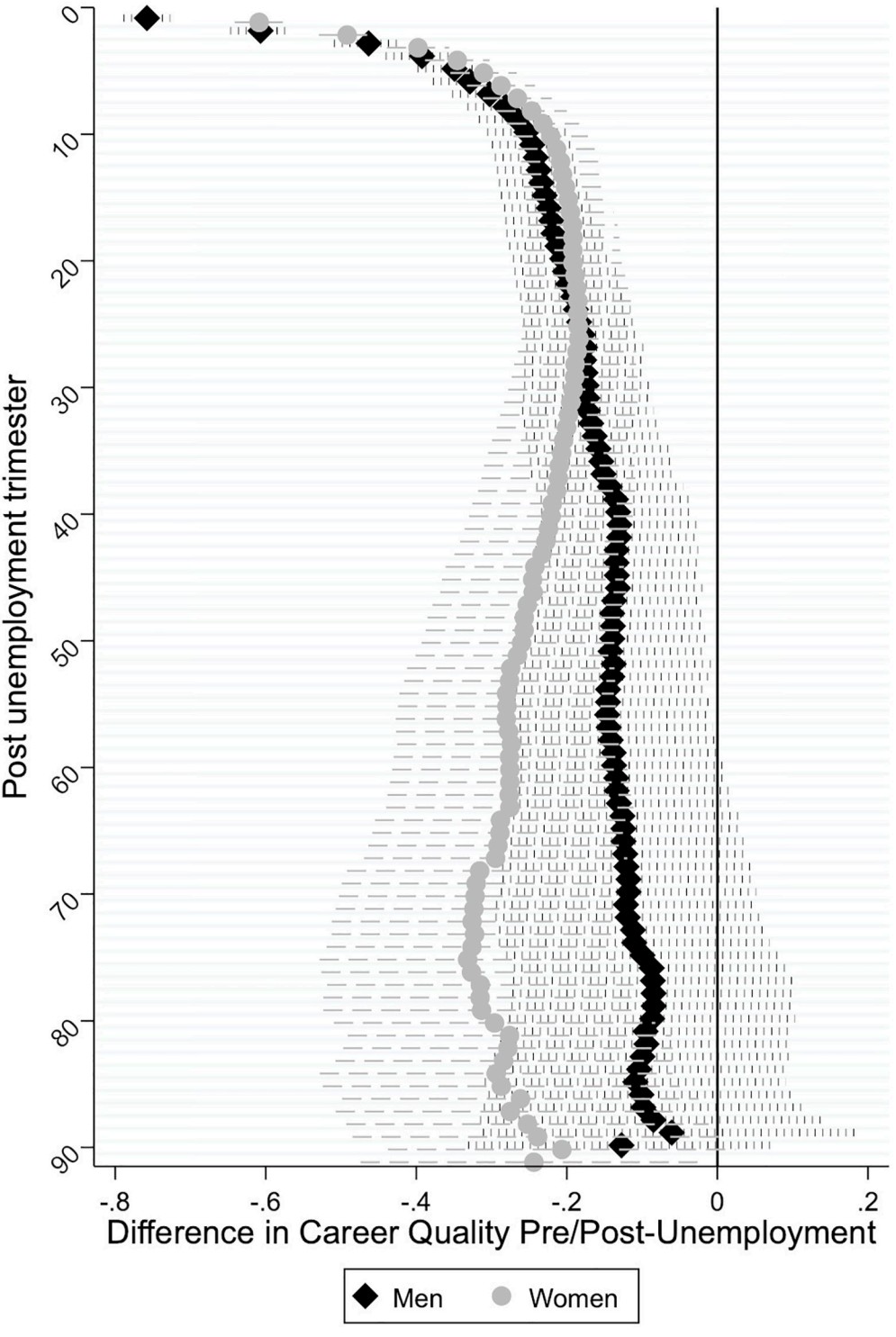

**Fig 1. Difference in career quality between pre and post unemployment period, by gender.**

## Post-unemployment career quality during childbearing and rearing periods

Next, we examine whether differences in the career quality pre and post-unemployment are wider for men during child-bearing (25–35) and rearing (36–45) ages than for their female counterparts (Hypothesis 2b). In addition, we test whether gender differences are statistically significant. Results of hybrid models are shown in S2 Table, which controls for changes in respondent's age, education, career quality prior to unemployment, and the GDP. Figs 2 and 3 plot "time" coefficients from those columns. Additionally, S2 Table shows the significance of gender differences, from a model including interactions between gender and the independent variables.

Fig 2 reveals a non-monotonic pattern of both career recovery and penalty among men and women who experience first unemployment during childbearing ages (25–35 years). Specifically, narrowing differences in the career quality between pre and post-unemployment take place in the initial periods following unemployment for both men and women in this age group indicating that both men and women find employment. However, men in this age group, show a relatively strong and short recovery period in the first year and a half following unemployment (as indicated by progressively smaller differences in the career quality between pre and post-unemployment period), after which their career quality remains stable and is not significantly different from their career quality before unemployment. The recovery among men suggests that they are able to find and retain employment. However, about 5 years after initial unemployment, men's recovery seems to stall and then revert. Although differences in the career quality pre and post-unemployment remain statistically non-significant (most likely due to the small sample size for this age group over the observed trimesters), the increasingly negative coefficients indicate a widening career penalty among men in this age category. For women in this age category, career penalties are more severe and persistent. Gender differences in time effects are statistically significant immediately after unemployment and around the first and second year following unemployment (i.e., trimesters 5 to 10). So far, these results are inconsistent with Hypothesis 2b.

Plotted coefficient estimates in Fig 3 clearly indicate that differences in career quality pre and post-unemployment are wider among men than women who experience their first unemployment during their childrearing ages (36–45 years). Specifically, although men in this age category experience a career recovery during the first 3 years, differences in the career quality compared to the pre-unemployment period widen significantly afterwards. Interestingly, women in this age group experience more moderate career penalties, as indicated by relatively smaller incremental differences in career quality compared to the pre-unemployment period. These results are consistent with Hypothesis 2b. Altogether, we find mixed support for Hypothesis 2b, that expected differences in the career quality pre and post-unemployment to grow wider for men during child-bearing (25–35) and rearing (36–45) ages than for their female counterparts.

## Age differences in post-unemployment career quality

Fig 4A (for men) and Fig 4B (for women) plot coefficients from a series of hybrid models that control for respondent's age, education, career quality before unemployment and GDP across five different age subsamples. In this way, we assume that each of the included variables has an age-specific effect on career quality. Coefficient estimates are shown in S3A and S3B Table.

Coefficients from hybrid models shown in S3A Table speak of differences in career quality between pre and post unemployment periods for individuals experiencing unemployment at older (36–64) and at younger ages (18–35), as we predicted in Hypothesis 3. Results indicate

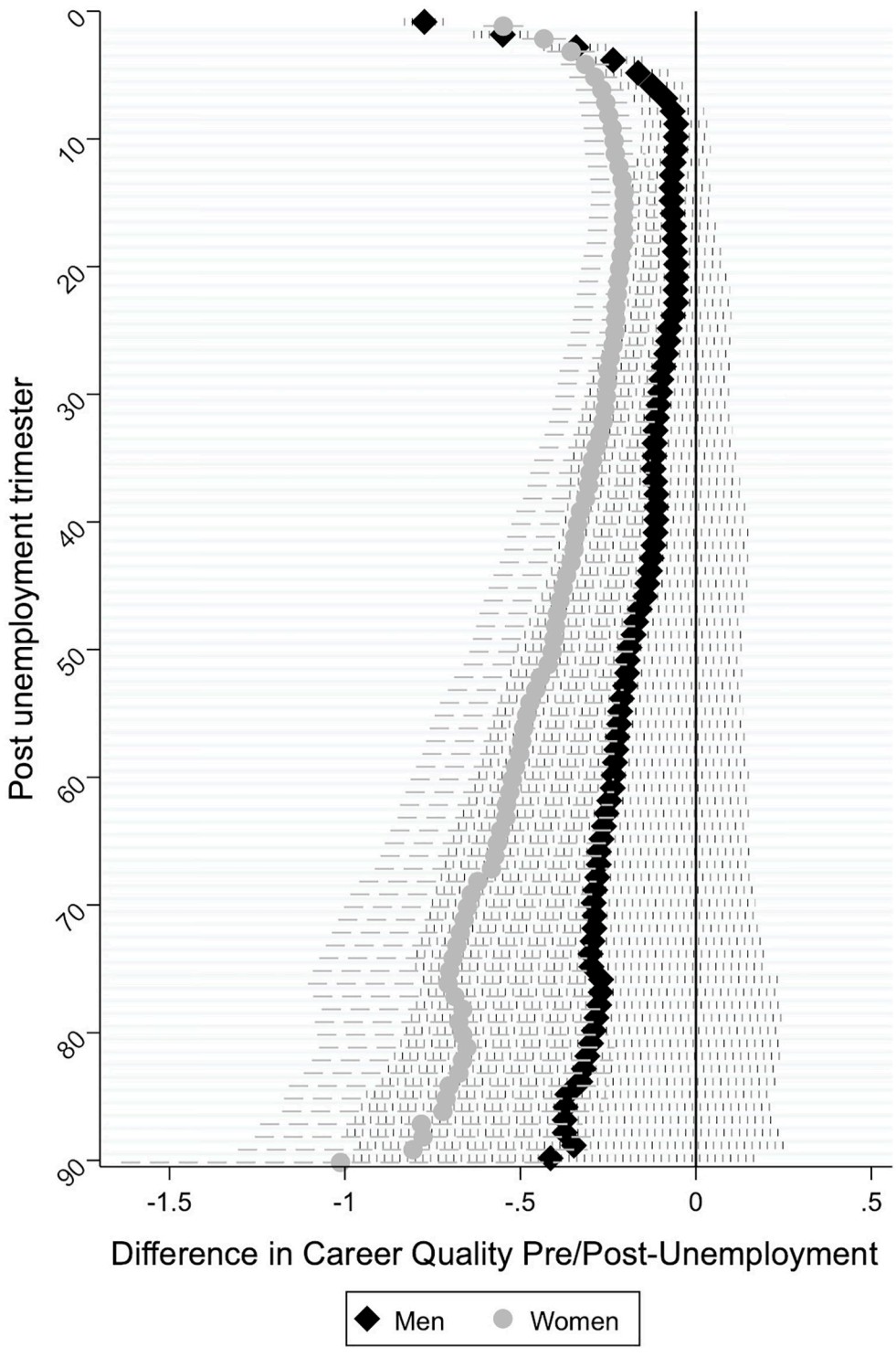

**Fig 2. Difference in career quality between pre and post unemployment period for those experiencing unemployment at childbearing ages (25–35 years).**

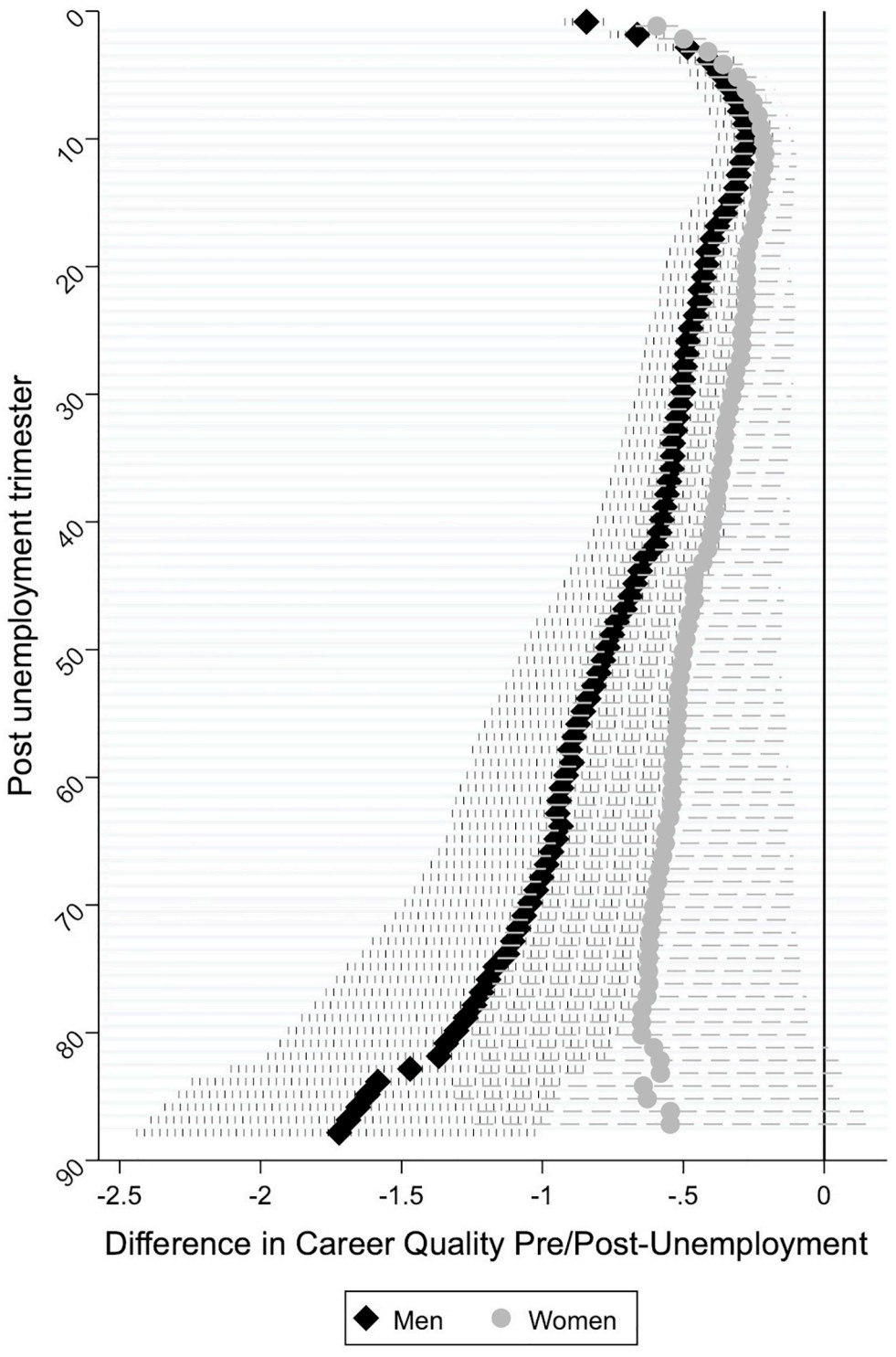

**Fig 3. Difference in career quality between pre and post unemployment period for those experiencing unemployment at childrearing ages (36–45 years).**

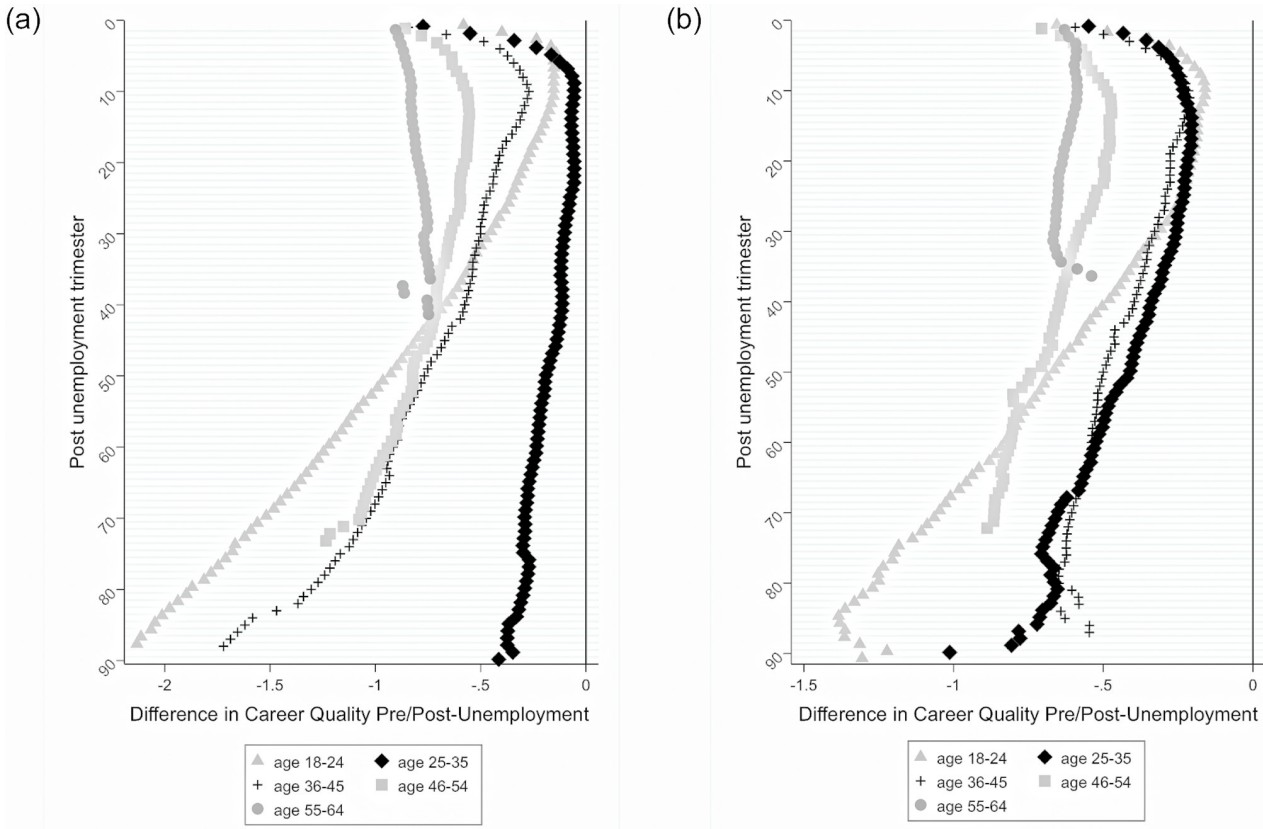

**Fig 4.** A. Difference in career quality between pre and post unemployment period for men experiencing unemployment at different ages. B. Difference in career quality between pre and post-unemployment period for women experiencing unemployment at different ages.

that differences in the level of career quality pre and post-unemployment, as well as the pace and pattern of recovery, differ for respondents experiencing unemployment at different ages. Men aged between 18–24 experience a recovery in their career quality in the first year after unemployment. After a year, the career recovery stabilizes and then reverts 2 to 3 years after the initial unemployment. Men aged between 25–35 also show a recovery in the first 5 trimesters (or 1.25 years) following unemployment. However, after this period, the career quality stabilizes at levels not significantly different compared to the pre-unemployment level. For men aged between 36–45, instead, the career quality recovers for a period of over two years, after which it starts decreasing rather sharply. Finally, men aged between 46–55 experience an increase in the career quality for about 3 years, followed by a (more moderate) decrease. Men aged between 55–64 at first unemployment show a recovery in the career quality in the first 6/7 years after unemployment, followed by a stabilization of career quality. These results suggest that for men: (a) careers can recover after an initial spell of unemployment, but that this recovery is (b) not distributed equally across the different age groups, with (c) men between the ages 25–35 least penalized. These results are consistent with Hypothesis 3.

Fig 4B shows that, similar to men, women of all different age groups experience an initial recovery in the career quality immediately after unemployment. The initial recovery in career quality is greater for younger women experiencing unemployment before the age of 35. This recovery in career quality slows down for women in this group, eventually stops after about 4 years following initial unemployment, and then starts to revert. For women that experience unemployment in the middle of their career, (i.e., between ages 36 and 55), we find evidence of

steep career penalties that stay relatively stable over the post-unemployment period, with some increase in the long term. Altogether, these results indicate that both younger and older women's post-unemployment careers are considerably and negatively impacted by unemployment. Career penalties follow a more dynamic pattern among younger women while older women experience more steady scars. These results are partially consistent with Hypothesis 3.

Finally, we formally tested our predictions that career quality trends following unemployment evolve differently by one's age at first unemployment (see S3A and S3B Table). For men, we find significant age differences compared to the youngest age group; those aged between 25–35 at first unemployment experience faster and stronger recovery; the evolution of career quality after the 4th year after unemployment is significantly different for the 25–35 age group, whose career quality does not deteriorate, contrary to the career quality of men 18–24, who experience a drop in career quality years after unemployment. Results show significantly stronger penalties for men experiencing unemployment at older ages in the first 8 years (31 trimesters) following unemployment. However, 13 years after initial unemployment these differences revert and become significantly wider for the youngest age group than the older age group (46–54 years). This provides mixed evidence for Hypothesis 3. For women, we find significantly different effects of time on career quality for those experiencing unemployment at older ages. Consistent with Hypothesis 3, our results indicate that older women (46–64) experience significantly stronger career penalties than younger women (18–24) for about 9 years after initial unemployment. While penalties remain rather constant for the oldest group of women, who are only observed for almost 9 years, for women 46–54 penalties decrease over the first 4 years but start to increase afterwards. For women experiencing unemployment earlier we see a similar pattern, characterized by an initial recovery in the first 3 to 4 years, followed by an increase in career penalties, which is the sharpest for women in the youngest age group.

## Discussion

In this study, we used detailed employment data from the GSOEP over the period 1984–2005 to investigate how career quality evolves over time after the occurrence of a first unemployment experience in Germany. We focused on Germany because of its relative generosity in the unemployment benefit institutions that are thought to shield people from the negative effects of unemployment. We paid careful attention to gender differences and timing of first unemployment. We theorized the endurance of unemployment effects on career quality using principles from the CAD perspective, expecting differences across gender and age of first unemployment. We used a new dynamic measure to quantify the quality of binary labor force sequences, distinguishing "good" (i.e., employment) from "bad" labor force status activities (i.e., unemployment and inactivity) and deployed a hybrid model that controls for respondents' stable unobserved differences.

Our first set of results support the proposition that differences in career quality between pre and post-unemployment narrow over one's career and disappear for men but not women. This "recovery" trend in career quality following unemployment applies to both men and women during the first 6 years after initial unemployment, after which it diverges, as women's career quality deteriorates, while men's continues to recover and then stabilizes. Against predictions from the CAD perspective, we find that such career recovery trends are not monotonic, but instead follow a non-linear trend such that the level of career quality first increases, then slows down and eventually stops or decreases (in the case of women). This finding indicates that unemployment carries a chronic scar particularly in the careers of German working women.

We also tested if the speed of recovery in career quality differs by gender and age at first unemployment. We found evidence that if first unemployment was experienced during their child-bearing ages (25–35), women's recovery in career quality is slower than for their male counterparts. These differences may reflect women's re-employment careers that become patchier and more fragmented after their time out of the labor market. However, we show weaker career penalties among women who experience initial unemployment during child-rearing ages (35–46) when compared to their male counterparts. Signaling effects and more generous benefit institutions related to women's employment histories could explain such differences. These results therefore indicate that the negative effects of unemployment are not distributed equally across gender and age groups. This finding is consistent with various studies reporting that youths in unstable jobs early in their careers suffer adverse labor market consequences over the course of their careers [2, 24].

Despite these contributions, our study has limitations that we hope future research will resolve. First, our analyses use employment alone as a measure of career success, without accounting for other employment characteristics. Although our study shows that women's careers stagnate after unemployment, they may also work in occupations of poorer quality and that pay less [15]. This would imply another type of penalty that reflects quality of jobs and employment contracts. The same may hold for age and occupational differences. Second, our measure of career quality was able to quantify binary sequences into "good" or "bad" sequences. This operationalization is useful to test binary predications from cumulative disadvantage theory and labor market segmentation theory, but it is not suited for career sequences that are more varied and with more than two categories in nature. Subsequent research could consider expanding the quality measure from a binary into a multi-categories measure. Third, due to its panel design, the GSOEP observed respondents during different time windows for different periods of time and at different stages of their careers. Consequently, the sample sizes (capturing the different trimesters) were larger in some subsamples than in others, with implications for the precision of our estimates. Analyses across more uniform samples could complement these results further. Finally, our study has been limited to the period prior to the Great Recession. Future research should investigate if and to what extent patterns of career recovery differ depending on economic fluctuations.

## Supporting information

**S1 Table. Hybrid model results.** Men and Women.
(DOCX)

**S2 Table. Hybrid model results: Subsamples of men and women ages 25–35 and 36–45.**
(DOCX)

**S3 Table.** A. Hybrid model results: Men experiencing unemployment at different ages. B. Hybrid model results: Women experiencing unemployment at different ages.
(DOCX)

## Author Contributions

**Conceptualization:** Anna Manzoni, Irma Mooi-Reci.

**Data curation:** Anna Manzoni, Irma Mooi-Reci.

**Formal analysis:** Anna Manzoni, Irma Mooi-Reci.

**Investigation:** Irma Mooi-Reci.

**Methodology:** Anna Manzoni, Irma Mooi-Reci.

**Project administration:** Anna Manzoni, Irma Mooi-Reci.

**Software:** Anna Manzoni, Irma Mooi-Reci.

**Validation:** Anna Manzoni.

**Visualization:** Anna Manzoni, Irma Mooi-Reci.

**Writing – original draft:** Anna Manzoni, Irma Mooi-Reci.

**Writing – review & editing:** Anna Manzoni, Irma Mooi-Reci.

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
