## [Decision Letter · Decision Letter 0]

16 Dec 2019

PONE-D-19-27271

The Cumulative Disadvantage of Unemployment: Longitudinal Evidence  Across Gender and Age at First Unemployment in Germany

PLOS ONE

Dear Associate Professor Manzoni,

Thank you for submitting your manuscript to PLOS ONE. After careful consideration, we feel that it has merit but does not fully meet PLOS ONE’s publication criteria as it currently stands. Therefore, we invite you to submit a revised version of the manuscript that addresses the points raised during the review process.

Both reviewers require substantial revisions; so, you should note that there is no guarantee for eventual publication after the revision round. If you choose to undertake the revisions, please try to address the reviewers' comments as fully as possible.

We would appreciate receiving your revised manuscript by Jan 27 2020 11:59PM. To enhance the reproducibility of your results, we recommend that if applicable you deposit your laboratory protocols in protocols.io, where a protocol can be assigned its own identifier (DOI) such that it can be cited independently in the future. For instructions see: http://journals.plos.org/plosone/s/submission-guidelines#loc-laboratory-protocols

We look forward to receiving your revised manuscript.

Kind regards,

Semih Tumen, PhD

Academic Editor

PLOS ONE

Journal Requirements:

**When submitting your revision, we need you to address these additional requirements:**

**Please ensure that your manuscript meets PLOS ONE's style requirements, including those for file naming. The PLOS ONE style templates can be found at http://www.plosone.org/attachments/PLOSOne_formatting_sample_main_body.pdf and http://www.plosone.org/attachments/PLOSOne_formatting_sample_title_authors_affiliations.pdf**

Reviewers' comments:

Reviewer's Responses to Questions

**Comments to the Author**

1. Is the manuscript technically sound, and do the data support the conclusions?

Reviewer #1: Partly

Reviewer #2: Partly

2. Has the statistical analysis been performed appropriately and rigorously? 

Reviewer #1: Yes

Reviewer #2: Yes

3. Have the authors made all data underlying the findings in their manuscript fully available?

Reviewer #1: No

Reviewer #2: Yes

4. Is the manuscript presented in an intelligible fashion and written in standard English?

Reviewer #1: Yes

Reviewer #2: Yes

5. Review Comments to the Author

Reviewer #1: The cumulative disadvantage of unemployment: Longitudinal evidence across gender and age at first unemployment in Germany

Comments to the Author

The present study examines a very interesting research question: to what extent are people who have experienced unemployment able to ‘restore’ their careers in the following years? The author(s) use data from the German Socio-Economic Panel (GSOEP) and hybrid models to shed light on this topic. Although the paper is both well written and innovative, there are a number of issues, listed below, that ought to be dealt with before the manuscript is ready for publication.

- The introduction would be further improved is the author(s) are more explicit on, first, exactly how the current study adds to the existing literature, and second, why this topic is of importance from a social policy perspective. Furthermore, some notes on what we can learn from the (unique?) German experience is warranted (either in the introduction or somewhere else in the manuscript).

- The author(s) should provide some more details on the “newly developed measure” that is applied. What is new? How is it an improvement? Are there any downsides?

- Why is the post-recessionary period excluded altogether from the analysis? In general, social scientist should strive to provide as ‘up-to-date’ knowledge as possible, and it therefore seems a bit strange to leave the years after 2005 out. The post-recession period could either be accounted for directly in the modeling (with e.g., calendar year or quarterly dummy variables), or the analyses could be run for both pre- and post-recession years. Alternatively, sound theoretical reasons for exclusion of post-2005 observations should be spelled out in more detail.

- To what extent is this study able to distinguish between human capital depreciation, on the one hand, and signaling theory on the other?

- There is a large literature on the so-called ‘scarring effects’ of unemployment, and several recent studies have used experimental data as well to examine the negative impact of gaps on the CV. These papers are of relevance for the current manuscript, and should probably be cited and discussed (at least to some extent). The experimental literature on age discrimination could be of relevance as well.

- We need some more details on the institutional setting/ contextual information, e.g., some notes on job search requirements, other potential income maintenance schemes, etc. for people experiencing unemployment.

- It might be a good idea to include a separate section on (i) previous research, and one section on (ii) theoretical perspectives.

- The hypotheses are formulated in a general fashion currently, and should either be re-formulated in a more specific manner, or perhaps rather be rephrased as (broad) research questions.

- How are the two central concepts “speed of recovery” and “career quality” operationalized/defined?

- What do we gain (and lose) from the sequence approach? What are the alternative strategies (if any), and why is the sequence approach the most preferable?

- The educational variable is currently categorical, but I would recommend to rather include 3 education dummies in the analysis (a more flexible modelling approach).

- I agree to the choice of parsimonious model specification, but the author(s) should spell out the reasons underlying this choice (e.g., multi-collinearity issues, fear of including so-called ‘bad controls’, etc.).

- How do you deal with observations with missing values?

- While reading the manuscript, I got the impression that the GDP variable is included as an individual level control, is that correct? If so, what does it tell us, how can it be interpreted? The variable will necessarily be identical for many observations per year.

- Some notes on the effect size(s) are warranted.

- Proper legends/labels and notes should be added to the figures in order to help readers understand what they actually tell us.

I would like to end by saying that I am very impressed by this work, and I wish you the best of luck both with the current revision and with your future research endeavors.

Reviewer #2: The study uses detailed employment data from the German Socio-Economic Panel Study over a long period of time after people’s first unemployment experience.

The study is very original in a three important aspects: (1) the long follow-up period after unemployment that is usually not considered in other research, (2) the measure of career quality as a time-changing variable that takes the labour market experiences since unemployment into account (most other studies focus just on wage at a point in time), (3) the focus on first unemployment whereas most other studies do not account for previous unemployment experience. Furthermore, I find it applaudable that gender and age differences are explicitly considered even thought I have some issues with respect to the research design you use to examine these gender-age variations (see below). However, I see a few fundamental problems with the research design of the paper in its current form. Furthermore, I have some other points with respect to the theoretical and methodological part of the paper. .

Fundamental problems

Hypothesis 1A and H1B state that the career quality of previously unemployed individuals will either decrease or increase over the post-unemployment period. However, it seems that this increase is measured not in comparison to the career quality prior to unemployment but in comparison to the state of unemployment itself (trimester 0). That makes H1A an impossible scenario, the career quality cannot further decrease compared to being unemployed.

There is an inconsistency in H2B and H3. Both hypotheses combined do not allow for variation in the effect of age for men. If the age groups 25-45 (H2b) and older workers (45+) (H3) all have slower expected recovery after unemployment then there is no age group that does not have slower expected recovery. Or did you mean workers aged 18-25 should have less strong unemployment scars? Also this is a problematic assertion because it (1) is inconsistent with previous literature as larger unemployment scars have been found for unemployment experienced at younger age groups in previous research and (2) the age group 18-25 is not represented in all labour market segments because the highly educated group is still studying in that age bracket in Germany.

I miss a comparison group of the people who do not experience unemployment. Therefore, your unexpected gender finding that contradicts previous research could be due to the fact that also women who do not experience unemployment are more likely than men to interrupt their careers and have spells of inactivity and non-work. This is a fundamental design problem of your study.

Other points

The theoretical part of the text is not always consistent. You start with elaborating on three reasons for labour market disadvantage after unemployment: human capital depreciation, signaling and labour market segmentation (p.6-7). The third theory is quite different in nature than the first two and this needs to be acknowledged. While the first two theories listed -human capital depreciation and signaling- are indeed explaining why the experience of unemployment would lead to worse outcomes, the labour market segmentation theory is merely stating that there are jobs with worse conditions in the economy but does not at all explain the reasons for unemployment scarring. To the contrary, labour market segmentation means that some jobs are more at risk for unemployment to begin with and hence the effect of unemployment is endogenous. Part of the issue is also that the first two theories are micro-theories while the segmentation theory is a macro theory. I suggest that the authors reconsider carefully how to build up the theoretical argument.

Furthermore, the human capital depreciation theory and signaling theory are often just mentioned as possible explanations for labour market disadvantage after unemployment without being explicitly tested. It would be good if you could indicate in the review of previous research evidence which previous studies actually found evidence for the mechanism and which studies merely mentioned the mechanism as a possible explanation without any attempt to test it.

It is unclear whether you managed to single out the first unemployment experience only for those respondents who had their first interview at age 18 or also for respondents who had their first interview at a later age. See last sentence page 11 and first sentences page 12 (line 251 onwards): but who retrospectively reported to have been continuously employed (either with the same or another employer) since they were 18 at their first interview

If you look at the first unemployment experience for all age groups then this should be reflected in the theoretical part and the hypotheses. Because unemployment is a different experience if it is a first time experience or a recurrent experience and it is unclear from your overview what the results of previous studies refer to.

While I like the concept and measure of career quality and I think it is highly original, it is a pity that all states of employment are treated as having the best level of career quality. You mention in your paper the large differentiation in types of jobs and labour market segmentation but that is an aspect that you’re not taking into account in the measure of career quality. Perhaps you could extend the measure by differentiating between types of employment?

Period effects. While you mention that you exclude the period of the most recent recession and you control for the year’s GDP/capita, it would be good to provide some information about how the economic climate might have affected the age of first unemployment experience and how this could have affected your results.

6. PLOS authors have the option to publish the peer review history of their article (what does this mean?). If published, this will include your full peer review and any attached files.

Reviewer #1: Yes: Kristian Heggebø

Reviewer #2: No

---

## [Author Response · Author response to Decision Letter 0]

21 Jan 2020

In the following pages, we explain the main changes to the text, following the order of the comments provided by the two reviewers. 

“A” denotes our answers. Page numbers in the answers refer to the new “clean” version of the manuscript, without track changes.

Reviewer #1: The cumulative disadvantage of unemployment: Longitudinal evidence across gender and age at first unemployment in Germany

Comments to the Author

The present study examines a very interesting research question: to what extent are people who have experienced unemployment able to ‘restore’ their careers in the following years? The author(s) use data from the German Socio-Economic Panel (GSOEP) and hybrid models to shed light on this topic. Although the paper is both well written and innovative, there are a number of issues, listed below, that ought to be dealt with before the manuscript is ready for publication.

- The introduction would be further improved is the author(s) are more explicit on, first, exactly how the current study adds to the existing literature, and second, why this topic is of importance from a social policy perspective. Furthermore, some notes on what we can learn from the (unique?) German experience is warranted (either in the introduction or somewhere else in the manuscript). 

A: We thank you for the positive comments on our study and suggestions that you make. Heeding your advice, at pages 4-5, we articulate more clearly that our study addresses 1) the recovery processes from unemployment; 2) the shape of recovery; and 3) how these relationships vary across age and gender groups. In addition, we conclude the new section on “Empirical Evidence on Unemployment Scarring” (pages 5-8) with a paragraph on how exactly our study contributes to extant research. We hope that in doing so we have addressed your concern.

- The author(s) should provide some more details on the “newly developed measure” that is applied. What is new? How is it an improvement? Are there any downsides? 

A: The career quality measure we apply was recently developed by Manzoni & Mooi-Reci (2018). In this paper, we provide a summary of the technical features of the measure at page 15-16. We refer to Manzoni & Mooi-Reci (2018) for details about the features and the mathematical derivations of the measure. As we explain in the text [page 4-5], this measure is innovative in that it quantifies the quality of a career in a holistic and dynamic way; by distinguishing between favorable (i.e., employment) and unfavorable labor force statuses (i.e., unemployment and inactivity) and by looking at their evolution over time, it captures the volatility of labor force trajectories and allowing us to test cumulative advantage/disadvantage processes. The use of this measure advances current scholarship on this issue by providing a dynamic measure to examining the evolution of unemployment effects. This is the first sequence-based measure that quantifies the overall quality of labor force outcomes, and thus the career recovery from unemployment. We outline how our study contributes to the current literature on unemployment scarring on page 8.

- Why is the post-recessionary period excluded altogether from the analysis? In general, social scientist should strive to provide as ‘up-to-date’ knowledge as possible, and it therefore seems a bit strange to leave the years after 2005 out. The post-recession period could either be accounted for directly in the modeling (with e.g., calendar year or quarterly dummy variables), or the analyses could be run for both pre- and post-recession years. Alternatively, sound theoretical reasons for exclusion of post-2005 observations should be spelled out in more detail. 

A: We agree with you that ideally, we use data that are: (1) rich on information about entries and exits from the labor market; (2) mature, meaning that they track entire individual careers; and (3) cover relatively stable economic conditions. The latter requirement ensures that our measure does not pick up effects related to fluctuating characteristics of the labor markets (e.g., number and type of jobs, firing costs, wage flexibility etc.). If both labor market characteristics and associated individual characteristics jointly determine the rate of recovery from unemployment, then our estimates will include correlated but unmeasured effects that emerge during the Great Recession. Thus, extending the time to cover the period after 2005 would only distort our analyses (even after inclusion of a period dummy for the pre/post-Recession period or running separate analyses in the period pre/post the recession). Against this background, we think that focusing on the period between 1984-2005 provides the best possible time period for our types of analyses. Future research should investigate if the patterns of recovery differ depending on an economic recession or not. We recognize this point in the revised discussion session.

- To what extent is this study able to distinguish between human capital depreciation, on the one hand, and signaling theory on the other? 

A: Our study is not able to distinguish between human capital and signaling effects. In explicating mechanisms driving recovery effects, our study focuses on discussing mechanisms that influence the direction and position of career development following initial unemployment. In doing so, we do not seek (or pretend) to investigate how human capital and signaling effects play out or which mechanism matters more and when. Studies seeking to distinguish between these effects already exist in the literature and are best answered using different types of data, such as data from audit or vignette studies. Heeding your advice, we now explicitly outline the specific scarring mechanisms from the perspective of the different theories. We also mention that combined together all three identified unemployment dimensions lead to downward career levels over time. 

- There is a large literature on the so-called ‘scarring effects’ of unemployment, and several recent studies have used experimental data as well to examine the negative impact of gaps on the CV. These papers are of relevance for the current manuscript, and should probably be cited and discussed (at least to some extent). The experimental literature on age discrimination could be of relevance as well.

A: This is a very good point. Following your suggestion, we have now added a new section that summarizes the extensive literature in this area. 

- We need some more details on the institutional setting/ contextual information, e.g., some notes on job search requirements, other potential income maintenance schemes, etc. for people experiencing unemployment. 

A: Details about the institutional setting of unemployment benefits in Germany have been discussed at page 11. We mention the eligibility criteria for accessing benefits, the duration and level of the benefits. We also discuss other income maintenance schemes, such as unemployment allowance, that is used upon exhaustion of unemployment benefits. 

- It might be a good idea to include a separate section on (i) previous research, and one section on (ii) theoretical perspectives. 

A: As mentioned above, using your suggestion we have now added a new section that summarizes the extensive literature in this area. 

- The hypotheses are formulated in a general fashion currently, and should either be re-formulated in a more specific manner, or perhaps rather be rephrased as (broad) research questions. 

A: Heeding your advice, we have slightly changed the formulation of the hypotheses to include the period prior to initial unemployment. We have also more clearly articulated how different theories derive different mechanisms underlying unemployment scarring and how this study looks at their combined effect in influencing the rate of recovery. Specifically, in our theoretical framework, we argue that human capital, signaling and segmented labor market theories all predict that unemployment effects will have negative effects on careers that become larger over time (i.e., cumulative disadvantages). In contrast, using findings from literature we then expect that institutions counteract such negative trends and thus lead to recovery. In doing so, we are not trying to differentiate between the effect of different mechanisms, which in turn would lead to very specific formulations of our hypotheses. Rather, we use these theories to explain the direction of effects over time that are formulated more broadly. 

- How are the two central concepts “speed of recovery” and “career quality” operationalized/defined? 

A: Career quality is in fact a central concept in our paper. As we describe in the section “Measures. Dependent variable: Career Quality”, this measure was developed in Manzoni & Mooi-Reci (2018), where details about the features and the mathematical derivations of the career quality measure are outlined.

As we explain in the text, career quality is a time-varying measure based on a non-alignment sequence technique (Elzinga 2003). The measure is developed based on a sequence of labor market states respondents go through following a first unemployment experience. We account for four major labor market states: employment, unemployment, inactivity, and retirement, measured on a monthly basis. The career quality measure accounts for the timing and recency of employment spells, which determine how fast one recovers from a first unemployment experience. The career quality increases with the number of employment episodes and their recency. In other words, the more frequent employment observations are and the more recent they are relative to the total sequence length, the higher the career quality will be. 

The evolution of career quality in the period following unemployment indicates whether and how fast one recovers from unemployment, that is, the speed of recovery. Our hybrid regression models look at the evolution of career quality over time; specifically, our models include time dummies measuring time after unemployment, which indicate the change in career quality compared to the career quality before unemployment (which is the reference category in the regression models). 

- What do we gain (and lose) from the sequence approach? What are the alternative strategies (if any), and why is the sequence approach the most preferable? 

A: The central question in our paper refers to whether and how fast individuals recover from unemployment. Such question calls for longitudinal data and for a holistic perspective measuring whole careers rather than as an instantaneous measure. We believe our approach best satisfies all those requirements.

- The educational variable is currently categorical, but I would recommend to rather include 3 education dummies in the analysis (a more flexible modelling approach). 

A: As you suggested, we operationalize education by including three dummies for education in our models. The three included dummies refer to: (1) low, (2) intermediate, and (3) high education. Low education level acts as the reference category. 

- I agree to the choice of parsimonious model specification, but the author(s) should spell out the reasons underlying this choice (e.g., multi-collinearity issues, fear of including so-called ‘bad controls’, etc.). 

A: Heeding your advice, on page 16, we explain that we have selected variables that we think are not endogenous to unemployment.

- How do you deal with observations with missing values? 

A: The only control variable with missing values is education (see table 1). We account for that by creating a separate category for missing (0.61% of cases). As we clarify in the text [page 14], we only include cases with pre-unemployment histories. 

- While reading the manuscript, I got the impression that the GDP variable is included as an individual level control, is that correct? If so, what does it tell us, how can it be interpreted? The variable will necessarily be identical for many observations per year. 

A: The GDP variable is an aggregate measure that is used as a proxy for economic growth in a given year. In our regression results, a positive coefficient for GDP will tell us that career quality is higher when GDP increases. However, we are not interested in interpreting the effect of GDP specifically. Instead, adding the GDP in the model allows us to control for business cycle effects that may bias our results. 

- Some notes on the effect size(s) are warranted. 

A: As indicated on page 19 of this revised manuscript, the coefficients for the trimester dummies for women’s and men’s career quality indicate the estimated differences in the career quality for each trimester following unemployment, relative to the trimester prior to unemployment (the reference category), all else equal. It is important to note that coefficient estimates that come closer to 0 indicate that differences the level of career quality between the pre and post-unemployment trimesters are no longer significant, indicating a sign of narrowing career penalties or what we call a ‘career recovery’. We have now also further commented on the size of the effects and specifically noted when effects are more or less sizable.

- Proper legends/labels and notes should be added to the figures in order to help readers understand what they actually tell us. 

A: We have now revised our figures to facilitate reading them.

I would like to end by saying that I am very impressed by this work, and I wish you the best of luck both with the current revision and with your future research endeavors.

We thank you very much for the detailed comments and suggestions, which we believe have improved the quality of our work. We hope that we have been able to address all your concerns. 

 

Reviewer #2: The study uses detailed employment data from the German Socio-Economic Panel Study over a long period of time after people’s first unemployment experience.

The study is very original in a three important aspects: (1) the long follow-up period after unemployment that is usually not considered in other research, (2) the measure of career quality as a time-changing variable that takes the labour market experiences since unemployment into account (most other studies focus just on wage at a point in time), (3) the focus on first unemployment whereas most other studies do not account for previous unemployment experience. Furthermore, I find it applaudable that gender and age differences are explicitly considered even thought I have some issues with respect to the research design you use to examine these gender-age variations (see below). However, I see a few fundamental problems with the research design of the paper in its current form. Furthermore, I have some other points with respect to the theoretical and methodological part of the paper. 

A: Thank you for the positive and encouraging comments you make about our paper. 

Fundamental problems

Hypothesis 1A and H1B state that the career quality of previously unemployed individuals will either decrease or increase over the post-unemployment period. However, it seems that this increase is measured not in comparison to the career quality prior to unemployment but in comparison to the state of unemployment itself (trimester 0). That makes H1A an impossible scenario, the career quality cannot further decrease compared to being unemployed.

A: Thank you for making this excellent point about the reference for comparison in our analyses of the evolution of career quality in the post-unemployment period. Based on your comment, we have redefined our comparison group as the trimester before first unemployment occurrence. Therefore, the time coefficients in our models now reflect the change in career quality in comparison to the career quality in the last trimester before first unemployment. In this way, the career quality could increase or decrease. 

There is an inconsistency in H2B and H3. Both hypotheses combined do not allow for variation in the effect of age for men. If the age groups 25-45 (H2b) and older workers (45+) (H3) all have slower expected recovery after unemployment then there is no age group that does not have slower expected recovery. Or did you mean workers aged 18-25 should have less strong unemployment scars? Also this is a problematic assertion because it (1) is inconsistent with previous literature as larger unemployment scars have been found for unemployment experienced at younger age groups in previous research and (2) the age group 18-25 is not represented in all labour market segments because the highly educated group is still studying in that age bracket in Germany.

A: Hypothesis 2b compares men 25-45 to women 25-45, predicting that men will recover more slowly than women. Hypothesis 3 compares older vs younger individuals, predicting that older individuals will recover more slowly than younger individuals. The two hypotheses are compatible as “slower recovery” is relative; in other words, men 25-45 may have slower recovery than women 25-45 but faster recovery than men 45+ for example. 

I miss a comparison group of the people who do not experience unemployment. Therefore, your unexpected gender finding that contradicts previous research could be due to the fact that also women who do not experience unemployment are more likely than men to interrupt their careers and have spells of inactivity and non-work. This is a fundamental design problem of your study. 

A: In this paper, we are interested in the evolution of career quality following a first unemployment experience. With our measure, continuously employed workers who have never been at risk of unemployment during their career would have a career success of 1 that would remain constant over time. Hence, for respondents who are continuously employed, there is no unemployment spell to recover from. Instead, we focus on the pool of the respondents who experiences unemployment to determine the rate of recovery. 

Other points

The theoretical part of the text is not always consistent. You start with elaborating on three reasons for labour market disadvantage after unemployment: human capital depreciation, signaling and labour market segmentation (p.6-7). The third theory is quite different in nature than the first two and this needs to be acknowledged. While the first two theories listed -human capital depreciation and signaling- are indeed explaining why the experience of unemployment would lead to worse outcomes, the labour market segmentation theory is merely stating that there are jobs with worse conditions in the economy but does not at all explain the reasons for unemployment scarring. To the contrary, labour market segmentation means that some jobs are more at risk for unemployment to begin with and hence the effect of unemployment is endogenous. Part of the issue is also that the first two theories are micro-theories while the segmentation theory is a macro theory. I suggest that the authors reconsider carefully how to build up the theoretical argument.

A: Heeding your suggestion, we have revised the theory section. We now conclude with a summarizing sentence that delineates the proposed mechanism(s) of unemployment scarring following from each theory. We now discuss how human capital theory points at the length of unemployment as the driving force behind unemployment scarring, while signaling theory points at the recency of unemployment as the mechanism that drives differences in hiring practices and hence influences the rate of recovery after unemployment. Segmentation theories on the other hand point at how previously unemployed are likely to end up in poor quality jobs that are highly correlated with employment instability, uncertainty and unemployment reoccurrence. The latter is another dimension of unemployment that influences the rate of recovery. 

Furthermore, the human capital depreciation theory and signaling theory are often just mentioned as possible explanations for labor market disadvantage after unemployment without being explicitly tested. It would be good if you could indicate in the review of previous research evidence which previous studies actually found evidence for the mechanism and which studies merely mentioned the mechanism as a possible explanation without any attempt to test it. 

A: In this revised version, we argue that all three unemployment dimensions (i.e., duration, recency and repetition) combined influence the direction of career quality downwards. Thus, we are not trying to measure the separate effects of each unemployment dimension. Instead, we are looking at how their combined effects influence the level of career quality compared to the period prior to first unemployment. 

 It is unclear whether you managed to single out the first unemployment experience only for those respondents who had their first interview at age 18 or also for respondents who had their first interview at a later age. See last sentence page 11 and first sentences page 12 (line 251 onwards): but who retrospectively reported to have been continuously employed (either with the same or another employer) since they were 18 at their first interview.

A: As we explain in the text [page 15] we do not limit our sample to respondents who were 18 years old at their first interview. Instead, we include respondents 18 or older who retrospectively reported to have been continuously employed (either with the same or another employer). In this way, we exclude the possibility that a respondent experienced unemployment prior to the interview. 

If you look at the first unemployment experience for all age groups then this should be reflected in the theoretical part and the hypotheses. Because unemployment is a different experience if it is a first time experience or a recurrent experience and it is unclear from your overview what the results of previous studies refer to.

A: Thank you for this suggestion. As mentioned above, in this revised version we highlight in the theory section which unemployment dimensions matter most according to different theories. We have also included a separate section on previous literature, which discusses the findings as these relate to different unemployment dimensions. We hope that in adding this new section we have been able to address your concern.

While I like the concept and measure of career quality and I think it is highly original, it is a pity that all states of employment are treated as having the best level of career quality. You mention in your paper the large differentiation in types of jobs and labour market segmentation but that is an aspect that you’re not taking into account in the measure of career quality. Perhaps you could extend the measure by differentiating between types of employment? 

A: As you rightly point out, and as we also acknowledge in the Discussion section, differentiating between the quality of different employment types would offer an additional refinement to our measure. However, for the objective of the current study, in which we are interested in investigating the quality of labor force statuses after initial unemployment, we think that the current measure provides an accurate estimate of career quality. 

Period effects. While you mention that you exclude the period of the most recent recession and you control for the year’s GDP/capita, it would be good to provide some information about how the economic climate might have affected the age of first unemployment experience and how this could have affected your results. 

A: Following your suggestion, we have briefly discussed the economic growth and unemployment in Germany during the period 1985-2005 (which covers our observation period). This is discussed at page 11.

We thank you for all your suggestions and comments that have been very valuable to us. We believe that in revising this manuscript we have been able to address all your concerns.

---

## [Decision Letter · Decision Letter 1]

12 May 2020

PONE-D-19-27271R1

The Cumulative Disadvantage of Unemployment: Longitudinal Evidence  Across Gender and Age at First Unemployment in Germany

PLOS ONE

Dear Associate Professor Manzoni,

Thank you for submitting your manuscript to PLOS ONE. After careful consideration, we feel that it has merit but does not fully meet PLOS ONE’s publication criteria as it currently stands. Therefore, we invite you to submit a revised version of the manuscript that addresses the points raised during the review process.

We have now received the second set of referee reports. You'll see below that the Reviewer #1 remains unconvinced about a few issues that s/he has raised in the first round. Please carefully address the remaining issues before the paper becomes publishable. In particular, please clarify why "the post-recessionary period is left out of the analysis" and also why "the current analytical strategy is better than the alternatives."

We would appreciate receiving your revised manuscript by Jun 26 2020 11:59PM. To enhance the reproducibility of your results, we recommend that if applicable you deposit your laboratory protocols in protocols.io, where a protocol can be assigned its own identifier (DOI) such that it can be cited independently in the future. For instructions see: http://journals.plos.org/plosone/s/submission-guidelines#loc-laboratory-protocols

We look forward to receiving your revised manuscript.

Kind regards,

Semih Tumen, PhD

Academic Editor

PLOS ONE

Reviewers' comments:

Reviewer's Responses to Questions

**Comments to the Author**

1. If the authors have adequately addressed your comments raised in a previous round of review and you feel that this manuscript is now acceptable for publication, you may indicate that here to bypass the “Comments to the Author” section, enter your conflict of interest statement in the “Confidential to Editor” section, and submit your "Accept" recommendation.

Reviewer #1: (No Response)

2. Is the manuscript technically sound, and do the data support the conclusions?

Reviewer #1: Partly

3. Has the statistical analysis been performed appropriately and rigorously? 

Reviewer #1: Yes

4. Have the authors made all data underlying the findings in their manuscript fully available?

Reviewer #1: No

5. Is the manuscript presented in an intelligible fashion and written in standard English?

Reviewer #1: Yes

6. Review Comments to the Author

Reviewer #1: Thank you so much for the revised version of the paper, you have definitely responded well to most of my concerns. However, there are still two issues that should be addressed properly before this paper is ready for publication.

1. I fail to see why the post-recessionary period is left out of the analysis. I find it difficult to accept that "stable economic conditions" is a requirement for these types of analyses. Economic busts and booms is a major part of the labor market in capitalistic societies, and it seems weird to just drop these observational points from the analysis altogether. At the very least, you should run the analysis for the pre- and post-recessionary period separately, and compare and discuss the differences and similarities. In fact, this would probably strengthen the study immensely and make it considerably more interesting.

2. It is still unclear exactly how and why the current analytical strategy is better than the alternatives. The study would actually profit from including a 'traditional' analysis (after first specifying what the traditional and poorer alternatives are) directly, and comparing the empirical results with the 'new' analytical strategy that the authors prefer. What is gained, and what is lost, with the new approach?

7. PLOS authors have the option to publish the peer review history of their article (what does this mean?). If published, this will include your full peer review and any attached files.

Reviewer #1: No

---

## [Author Response · Author response to Decision Letter 1]

2 Jun 2020

Thank you for the positive comments on our manuscript. We are glad to hear that – with exception of two final issues - we addressed all your concerns both appropriately and rigorously. We are grateful for the very useful feedback and suggestions, which helped us further improve our work. Please find below a set of notes summarizing our responses to your comments. Your original comments are reproduced in italics font, while our answers are preceded by “A”.

I fail to see why the post-recessionary period is left out of the analysis. I find it difficult to accept that "stable economic conditions" is a requirement for these types of analyses. Economic busts and booms is a major part of the labor market in capitalistic societies, and it seems weird to just drop these observational points from the analysis altogether. At the very least, you should run the analysis for the pre- and post-recessionary period separately, and compare and discuss the differences and similarities. In fact, this would probably strengthen the study immensely and make it considerably more interesting.

A: In what follows, we provide additional evidence refuting a pre/post 2005 type of analysis. We start by including 10 additional waves of the GSOEP data extending our sample to 2015. Next, we estimate a series of fixed-effect regression panel models predicting the effect of time since first unemployment on career quality, separately across two periods. That is, 10 years before and 10 years after 2005. The coefficient estimates for the effect of time since first unemployment on the level of career quality across these two periods are shown in Figure 1 below. In addition, we also estimate a model that merges both periods before and after 2005 and controls for the time pre/post 2005. This reflects the size of the bias in the coefficients had we included data post 2005.

Figure 1. Pre/Post Analysis (see attached .doc)

Figure 1 reveals two important findings. First, effects of time since initial unemployment on career quality are considerably greater after 2005 than before 2005. For example, in the first year after unemployment the size of the coefficient in the post 2005 model is double the size of the coefficients in the pre 2005 model. We observe major differences in the coefficient estimates pre/post 2005 over a 10-year post unemployment time span (as evidenced by non-overlapping confidence intervals). Second, the recovery from unemployment post 2005 is considerably slower. These two key results reflect our concern that extending the analyses beyond 2005 will inflate and thereby bias our coefficient estimates downwards. 

One way to account for this bias, would be to apply a difference-in-difference approach. However, because the recession affected all workers, we are left with no control group to permit such an approach. The absence of a proper control and treatment group (in addition to a period before/after 2005) prevents us from using a difference-in-difference design to separate out recession-specific from individual-specific effects. The approach that you are suggesting – i.e., running separate analyses in the period pre/post 2005 and discussing similarities and differences in the two periods – is problematic for another reason. 

Substantively, it would generate a fundamentally different paper addressing different research questions compared to the ones we set up to address. In our original manuscript, we did discuss the issue around the observation window (i.e., lines 639-641), listing this as a potential avenue for future research. Overall, we believe that we have provided both empirical evidence and substantive arguments to justify our decision to investigate careers over a 22 year period (instead of extending analyses beyond 2005). We hope that you will be sympathetic to our efforts to respond diligently to your comments.

It is still unclear exactly how and why the current analytical strategy is better than the alternatives. The study would actually profit from including a 'traditional' analysis (after first specifying what the traditional and poorer alternatives are) directly, and comparing the empirical results with the 'new' analytical strategy that the authors prefer. What is gained, and what is lost, with the new approach?

A: Previous studies on the effects of unemployment – which we assume you refer to as “traditional approaches” have usually focussed on one unemployment dimension (i.e., occurrence or spell of previous unemployment) on outcomes such as earnings, employment, health or fertility of displaced workers. While valuable, this approach fails to capture labor force status changes over the course of one’s career, and how these in turn shape people’s employment trajectories and their level of career success in the future. Research on career ‘turbulence’ or ‘complexity’ based on a sequence analytical approach exist, but these measure the ‘volatility’ of labor force changes rather than the quality of these changes. 

In order to permit analysis of the degree and speed of recovery from an initial unemployment experience, we believe that a holistic perspective is necessary, which accounts for multiple dimensions of unemployment (e.g., occurrence, duration, frequency) simultaneously. As we outline in the Introduction, our sequence analytic approach provides such a holistic and multidimensional perspective. It quantifies the quality of labor force status trajectories in a dynamic way, capturing the volatility of labor force trajectories, their evolution since the occurrence of an initial unemployment spell and most importantly the quality of these transitions. In doing so, it advances “traditional” approaches that are based on instantaneous measures and a single dimension of unemployment. We also believe that by quantifying the quality of career success we make an important contribution to the sequence analytical approach. These issues are discussed and summarized in lines 142-158 of our manuscript. Nevertheless, following your advice we have now revised the Introduction and discuss the knowledge gaps more clearly (i.e., lines 71-79).

Thank you very much for your helpful suggestions and encouragement! We hope that we have been able to address your concerns.

---

## [Editor Report · Decision Letter 2]

3 Jun 2020

The Cumulative Disadvantage of Unemployment: Longitudinal Evidence  Across Gender and Age at First Unemployment in Germany

PONE-D-19-27271R2

Dear Dr. Manzoni,

We’re pleased to inform you that your manuscript has been judged scientifically suitable for publication and will be formally accepted for publication once it meets all outstanding technical requirements.

Kind regards,

Semih Tumen, PhD

Academic Editor

PLOS ONE

---

## [Editor Report · Acceptance letter]

12 Jun 2020

PONE-D-19-27271R2 

The Cumulative Disadvantage of Unemployment: Longitudinal Evidence  Across Gender and Age at First Unemployment in Germany 

Dear Dr. Manzoni:

I'm pleased to inform you that your manuscript has been deemed suitable for publication in PLOS ONE. Congratulations! Your manuscript is now with our production department. 

Kind regards, 

on behalf of

Professor Semih Tumen 

Academic Editor

PLOS ONE